# Slaughterhouse Visual and Palpation Method for Estimating the Economic Damage of Porcine Proliferative Enteropathy (PPE)

**DOI:** 10.3390/ani13030542

**Published:** 2023-02-03

**Authors:** István Szabó, István Makkai, Péter Máté, Tamás Molnár, Hanny Swam, Stephan von Berg, Derald J. Holtkamp, Róbert Glávits, István Szabó, László Ózsvári, László Búza

**Affiliations:** 1Enviroscience, Ltd., H-2115 Vácszentlászló, Hungary; 2Intervet Hungaria Kft., H-1095 Budapest, Hungary; 3MSD CDS, 5831 AN Boxmeer, The Netherlands; 4Intervet Deutschland GmbH, 85716 Unterschleissheim, Germany; 5Department of Veterinary Diagnostic and Production Animal Medicine, College of Veterinary Medicine, Iowa State University, Ames, IA 50011, USA; 6Autopsy Path Kft., H-1147 Budapest, Hungary; 7Department of Veterinary Forensics and Economics, University of Veterinary Medicine, H-1078 Budapest, Hungary

**Keywords:** visual and tactile method, slaughterhouse, *Lawsonia intracellularis*, economical losses

## Abstract

**Simple Summary:**

The profit production of the pig industry is fundamentally influenced by the functioning of fatteners’ digestive systems. Diseases that damage any section of the intestine reduce the intensity of nutrient absorption and worsen the economy of production. Ileitis is one of the most important and widely spread enteric diseases. Economic losses due to ileitis have been estimated at USD 4.65 per fattening pig, with American pig farmers losing USD 56.1 million annually. Currently, as there is no monitoring tool at the slaughterhouse to evaluate these losses, it has become necessary to develop an ileitis monitoring tool that is inexpensive, simple, fast, sensitive and provides immediate results. The method developed is similar to lung lesion scoring tools and can be performed at slaughterhouses. The results of the procedure are correlated with the results of other laboratory diagnostic tests for ileitis. The results can provide an immediate interpretation of the status of ileitis.

**Abstract:**

Background: Ileitis is a wasting disease of pigs. Clinical symptoms are diarrhea in growing pigs, wasting and reduced performance. Ileitis is ubiquitous in pig producing countries all around the world. It is estimated that the economic losses caused by the disease are USD 4.65 per fattening pig, and American pig farmers lose USD 56.1 million annually. It has become necessary to develop a slaughterhouse ileitis monitoring method that is simple, feasible to perform at modern slaughter lines, leads to immediate results and is cost effective. The practical experiences of applying the method are presented below. Methods: Our studies were performed on pig herds and slaughterhouses in Central European countries (Hungary, Romania, Poland, Croatia, and Slovakia). Experiences were evaluated based on visual and palpation. The results of our investigations were evaluated by a scoring method. Authors made histological and immunohistochemical examinations of investigated ileums. The hypothetic economic losses due to the disease in each farm were determined by estimating the loss of profit according to Holtkamp’s presentation in 2019. Results: The essentials of the method we have developed are: it can be performed during slaughterhouse processing, it does not interfere with or make it impossible to carry out normal technological processes, and the results of the procedure are correlated with the results of other laboratory diagnostic tests for ileitis (histology, immunohistochemistry, herd serology, fecal PCR). It is noteworthy that the results of the method can be used to immediately estimate the impact of *Lawsonia intracellularis* infection on the performance of the herd from which the slaughter animals come. Conclusion: Using the results of the slaughtered pigs’ visual and tactile examination at the slaughterhouse, the magnitude of the loss caused by *Lawsonia intracellularis* infection can be estimated quickly and accurately, and the return on investment of the strategy to be applied can be accurately planned.

## 1. Background

Ileitis or porcine proliferative enteropathy (PPE) is a wasting disease of pigs. PPE includes all forms of disease characterized by thickening of the small intestine, including the ileum, due to the proliferation of cells, even in the colonic mucosa. Clinical symptoms are diarrhea, wasting and reduced performance (decreased average daily weight gain, ADG) in growing pigs. In fattening and breeding animals, proliferative hemorrhagic enteropathy (PHE) may result in the discharge of blood in the feces. The disease can also occur in a chronic form in fattening herds, accompanied by mild diarrhea and poor weight gain (porcine intestinal adenomatosis, PIA) [1,2].

PPE is ubiquitous in pig producing countries all around the world. Holtkamp et al. estimated the economic losses caused by the disease at USD 4.65 per fattening pig, with American pig farmers’ losses at USD 56.1 million annually [3].

Diagnosis of the disease is possible based on pathological and histopathological lesions. Immunohistochemistry or PCR may be required for causal diagnosis.

Jensen et al. (1997) reported on the monitoring of porcine enteric disease in a slaughterhouse scoring system similar to that regarding mycoplasmosis [4]. In their studies, the ileac sections of 39 fattening pigs slaughtered were examined and graded by visual and palpation monitoring. According to their results, it is possible to monitor and classify the intestines after slaughter visually and by palpation based on the rigidity and thickness of the ileum. In their view, it is essential to demonstrate typical proliferation or to detect *Lawsonia intracellularis* bacteria.

In Hungary, a slaughterhouse belonging to a significant pig integrator realized that the number of small intestines suitable for further processing decreased significantly (by almost 60–70%) during the production process due to inflammation of the small intestine, rupture, and significant narrowing of the intestinal lumen. Histological examinations confirmed the intracellular presence of *Lawsonia intracellularis* in samples showing macroscopic lesions [5].

Based on the above, it has become necessary to develop a slaughterhouse ileitis monitoring method that is simple, feasible to perform at modern slaughter lines, leads to immediate results and is cost effective. It should be doable during slaughterhouse processing without disrupting the slaughter processes themselves. The results of the method should correlate with the results of other laboratory diagnostic tests for ileitis (histology, immunohistochemistry, herd serology, fecal PCR). It is important that the results of the method can be interpreted immediately in terms of the performance of the herd from which the animals arrived for slaughtering. The practical experiences of applying the method are presented below.

## 2. Materials and Methods

### 2.1. Herds Tested and Animals Slaughtered

Our studies were performed on pig herds and slaughterhouses in Central European countries (Hungary, Romania, Poland, Croatia, and Slovakia).

#### 2.1.1. Hungary

Between October and December 2020, we performed ileum testing on 463 slaughter pigs from 13 large-scale pig farms in Hungary. Of these farms, 7 were farrow-to-slaughter type breeding ones while 6 were fattening units. *Lawsonia intracellularis* was confirmed in all breeding farms and in 3 out of 6 fattening farms by serological tests before slaughterhouse monitoring started. In 12 of the farms, no preventive or control measures were applied regarding ileitis. One farrow-to-slaughter farm used Porcilis^®^ Lawsonia for more than half a year. At all slaughterhouses, the examinations included the visual inspection and palpation of the ileum section of the intestinal tract of fattening animals from each farm.

#### 2.1.2. Romania, Poland, Croatia, Slovakia

In February–March 2021, we examined the ileum of 383 pigs arriving at the slaughterhouse using the scoring method developed by the methods described in this manuscript. Pigs originated from 4 Romanian, 4 Polish, 3 Croatian and 3 Slovakian swine units. The presence of *Lawsonia intracellularis* in all farms was earlier confirmed by serological tests. One Romanian farm used PorcilisÂ^®^ Lawsonia to control ileitis.

We sampled 25–53 slaughtered pigs per farms. This corresponds to an incidence level of 10% at a prevalence level of 95%. Based on this, we could highlight the group of animals which scored “3” in the visual inspection and palpation investigation, representing serious cases, and see if the herd mortality due to ileitis was correlated.

### 2.2. Ileitis Slaughterhouse Monitoring Method (EnteriPig)

Experiences were evaluated based on visuals and palpation. During the evaluation, the following system was developed:

#### Visual Inspection and Palpation Evaluation

**0** = no perceptible lesion, no hard tact formula in the intestine, no thickening at the ileocecal intestinal cross section, no color change visible from the outside;

**1** = one or two palpable hard tact 0.5–1 cm formula in the intestine, slightly noticeable folds in the intestine, slight increase in intestinal wall thickness;

**2** = multiple, hard tact oval formula in the intestine, pronounced folds in the intestine, thickening at the ileocecal intestinal cross section, discoloration;

**3** = blood content as described in the 2 evaluations and already visible from the outside (serosa).

### 2.3. Post-Mortem Examination of the Mucosa of the Ileum of Slaughter Pigs

After visual inspection and palpation, we made incisions into the intestinal section and recorded the color of the mucosa, the smoothness of the mucosal folds, the palpation of any lesions found, and the quality (color, texture) of the contents. The results of our investigations were evaluated by a scoring method.

#### Test 1: Examination of the Ileum Mucosa at the Slaughterhouse

##### Evaluation of Ileum Mucosa

**0** = no lesions, normal mucosa, mucosa does not stand out from its surroundings, no swollen lymph follicles;

**1** = ulcerated lymph follicles with a black surface, the mucosal pleats can be smoothed;

**2** = pronounced, black, ulcerative lymph follicles extending to the entire lymphatic follicle, mucosal pleats area cannot be smoothed;

**3** = as described in 2 evaluations, as well as severe bleeding lesions on the intestinal mucosa.

### 2.4. Serological Examination

Serological testing of blood samples from different pig age groups was performed on each farm. *Lawsonia intracellularis* antibodies were determined by the SVANOVIR^®^ L. in-tracellularis Ileitis-Ab ELISA system.

### 2.5. PCR Investigations

The ileum content and/or mucosal scraping was sampled for PCR analysis. In some cases, we even collected a pooled fecal sample before slaughter from groups of animals that were later slaughtered. PCR analysis was performed by BactoReal^®^
*Lawsonia intracellularis* Standard quantitative PCR.

### 2.6. Histopathological and Immunohistochemical Examination

From the examined ileum, an approximately 1-square-centimeter piece was excised and fixed in formalin for histological and immunohistochemical examination.

For histopathological examinations, samples were fixed in 10% buffered formaldehyde solution. The samples were then embedded in paraffin. Sections were taken with a microtome and then placed on a slide. Plates were stained with hematoxylin-eosin and examined by light microscopy.

Immunohistochemistry was performed using BIO-X Diagnostics (anti-lawsonia intracellularis monoclonal antibody (BIO 323) reagent for indirect immunofluorescence or immunoperoxidase assay) for detecting *Lawsonia intracellularis* in tissue sections.

### 2.7. Statistical Analyses

We applied Cohen’s kappa coefficient (κ) analyses between visual inspection and palpation method and mucosal membrane results at slaughterhouse.

### 2.8. Estimation of the Economic Damage of the Disease Based on the Visual and Tactile Monitoring of the Slaughterhouse

The economic loss due to ileitis in each farm was estimated as the loss of profit based on a study to estimate the economic impact of ileitis in the finishing phase of production conducted by Holtkamp, 2019 [6]. Using a production and economic model, Holtkamp estimated the profit on a EUR per pig marketed basis for pigs unaffected by ileitis and for pigs affected by ileitis for a lower and upper bound of the severity of the disease. An average start weight of 22 kg and a fixed time of 115 days on feed was used in each scenario. Therefore, as ADG changed, the average market weight changed accordingly. A market pig price of EUR 1.55 per kg and a feed price of EUR 167 per ton were used in the production and economic model. The estimated profit for pigs unaffected by ileitis was EUR 16.58 per pig, compared to EUR 11.32 and EUR 1.67 per pig for the upper and lower bounds of severity, respectively, in pigs affected by ileitis. We also considered that very severe acute bleeding (PHE) ileitis at the slaughterhouse is associated with outstanding mortality during the growing period.

To estimate the economic losses due to ileitis in each farm, the reported profit from Holtkamp, 2019 was assigned to animals according to the ileum score at the slaughterhouse based on visual inspection and palpation. Pigs with an ileum score of “0” were deemed unaffected, those with a value of “1” were deemed affected at the lower bound, and those with a value of “2” or “3” were deemed affected at the upper bound. The % incidence of individuals with acute PHE lesions (value “3”) was used to describe mortality during fattening. There are a very limited number of data about how mortality increases with acute PHE [6], but since symptoms are visible during slaughterhouse examination the authors considered the production of these animals to be a total loss.

The loss of profit was calculated as the difference between the profit of unaffected pigs and those affected by ileitis and was estimated for each farm based on the prevalence of pigs in each category of ileum scores.

## 3. Results

### 3.1. Ileitis Slaughterhouse Monitoring Method (ENTERIPIG)

#### 3.1.1. Results of Tests Performed on Slaughter Pigs Produced in Hungarian Swine Farms

Slaughterhouse scoring results are summarized in Table 1. We found that we had fewer “0” accurate results for mucosal and intestinal analysis than for visual inspection and palpation examination of the same intestinal tract. Based on the palpation examinations, 26.8% of animals were scored “0”, 46.9% were scored “1”, 25.9% were score “2” and 0.4% were scored “3”.

#### 3.1.2. Results of Tests Performed on Pigs for Slaughter Produced in Pig Holdings in Romania, Poland, Croatia, and Slovakia

In February–March 2021, 383 pig ilea were examined from four Romanian, four Polish, three Croatian and three Slovakian swine farms using the scoring method we developed. The presence of *Lawsonia intracellularis* in all herds was confirmed by serological tests earlier.

Based on the palpation examinations, 39.0% of animals scored “0”, 45.0% scored “1”, 14.0% scored “2” and 1.0% scored “3”. We found “3”-rated intestines in 1% of cases. We were not able to make intestine mucosa investigations in Poland and Romania because of a lack of a comfortable place. The results of mucosal examinations after cutting up the intestines from the Croatian and Slovak farms correlated with the palpation method (Table 2).

Results of Cohen’s kappa coefficient (κ) analyses between visual inspection and palpation method and mucosal membrane results at the slaughterhouse are shown in Table 3. We observed significant agreement between the two methods.

### 3.2. Histopathological and Immunohistochemical Examinations

#### 3.2.1. Farms in Hungary

Of the 463 ilea examined, 190 (41%) were collected for histological and immunohistochemical examination. The results of these studies are summarized in Table 4. Of the samples graded “0”, “1” and “2” by slaughterhouse visual inspection and palpation methods, histological examination showed severe or very severe (grade 2, grade 3) epithelial exfoliation in 6%, 32% and 83%, and significant and pronounced proliferation of inflammatory cells in 11%, 45%, 80%, respectively. The incidence of other histopathological lesions (epithelial cell proliferation in crypts, edema, bleeding) was significantly higher in macroscopic grade 2 samples than in non-abnormal indicators. Histological examinations revealed no fibrosis and rare (4 cases out of 190) lymphoid hyperplasia. 

We calculated Cohen’s kappa coefficient (κ) between visual and palpation method at slaughterhouse and found histopathological lesion. The result is shown in Table 5. We found very strong, significant agreement between palpation at slaughterhouse and both the exfoliation and inflammatory cell infiltration lesions of histopathological investigations.

During the immunohistochemical examinations, the presence of *Lawsonia intracellularis* was detected in only one (2%) of the samples rated “0” by the slaughterhouse visual and palpation method, while six (10%) of the “1” rated and twelve (15%) of the “2” rated were positive.

We also calculated the Cohen’s kappa coefficient (κ) between visual and palpation method at slaughterhouse and *Lawsonia intracellularis* detected by immunohistochemistry (Table 5). We observed the highest significant (*p* < 0.05) agreement (61%) between visual and palpation method at slaughterhousewith score “2” and the immunohistochemistry results.

#### 3.2.2. Farms in Romania, Poland, Croatia, and Slovakia

Of the 383 ilea examined at slaughterhouses, 92 (24%) were collected for histological and immunohistochemical examination. The results of these studies are summarized in Table 6. Of the samples graded “0”, “1” and “2” by slaughterhouse visual and palpation methods, histological examination showed severe or very severe (grade 2, grade 3) epithelial exfoliation in 0%, 14% and 41%, and significant and pronounced proliferation of inflammatory cells in 0%, 10%, 32%, respectively. Epithelial cell proliferation in the crypt were found in three cases of grade “2” samples. There was no incidence of other histopathological lesions (edema, hemorrhage, lymphoid hyperplasia, and fibrosis).

We calculated Cohen’s kappa coefficient (κ) between visual and palpation method at slaughterhouse and the histopathological lesions. Results are shown in Table 7. We found very strong, significant agreement between palpation at slaughterhouse and both the exfoliation and inflammatory cell infiltration lesions of histopathological investigations.

During the immunohistochemical examinations, the presence of *Lawsonia intracellularis* was detected in three (6.82%) of the samples rated “2” by the slaughterhouse visual inspection and palpation method.

We also calculated the Cohen’s kappa coefficient (κ) between visual and palpation method at slaughterhouse and *Lawsonia intracellularis* detected by immunohistochemistry (Table 7). We did not observe significant (*p* < 0.066) agreement (55.4%) between visual and palpation method at slaughterhousewith the score “2” and the immunohistochemistry results.

### 3.3. Serological Examination

Antibodies against *Lawsonia intracellularis* were detected by herd-level age group testing in all tested farms. Based on this, the examined swine farms were classified as infected with ileitis (Table 1 and Table 2).

### 3.4. PCR Examinations

*Lawsonia intracellularis* PCR analysis of fecal samples revealed that there were non- vaccinated farms in which no pathogens could be detected in at least one sample. We unfortunately did not have samples from vaccinated farms (Table 8).

Samples from farms in Romania and Poland could not be subjected to PCR due to too long delivery times.

We took 32 samples of mucosal membrane, and ileum content for PCR investigation.

Two of the Slovakian samples were *Lawsonia intracellularis* positive, but below the limit of detection. We had the same result in one case of 14 samples according to Croatian pigs, and four of 14 samples had 1.0, 1.3, 1.5 and 1.5 10Log Copies/µL.

### 3.5. Estimation of the Economic Damage of the Disease Based on the Visual and Tactile Monitoring of the Slaughterhouse

#### 3.5.1. Farms in Hungary

Eleven of the thirteen Hungarian farms realized only 50–70% of the maximum achievable profit if all of the pigs had been unaffected by ileitis. (Table 9)

It should be emphasized that where the bleeding form of the disease also appears (G) it makes fattening practically economically ineffective (Table 9).

Farm (M), which used Porcilis^®^ Lawsonia i.m. vaccination against ileitis, showed a 15% reduction in profit from the achievable profit (i.e., no affected pigs in the herd).

#### 3.5.2. Swine Farms in Romania, Poland, Croatia, and Slovakia

Up to a 100 percent reduction in profit from the achievable profit (i.e., no affected pigs in the herd) was found in one farm (K). Two out of fourteen farms showed 20–42% of the calculated achievable profit, in nine farms out of fourteen 62–76% of the calculated achievable profit, and in one (farm B) out of fourteen cases 82 % of the calculated achievable profit was realized.

It should be emphasized that, where the bleeding form of the disease also appears (Farms A, D, K), this makes fattening practically unprofitable or of very low profitability.

At the farm using the Porcilis^®^ Lawsonia vaccine against ileitis (C), the loss of profit was 4%, so 96% of the profits achievable compared to the ‘Lawsonia-free’ status were realized (Table 10).

## 4. Discussion

Among non-notifiable infectious diseases that threaten the profitability of modern pig farming, those causing respiratory symptoms (PRDC) and gastrointestinal damage play a prominent role. In the latter disease group, in the recent period, the incidence of swine dysentery damage has been reduced through the production of free herds and compliance with internal disease control rules.

However, another group of gastrointestinal diseases increased: proliferative enteropathy (PPE) in pigs in the form of chronic disease (porcine intestinal adenomatosis (PIA) in pigs, in the form of acute, severe clinical symptoms (porcine hemorrhagic enteropathy in pigs, PHE)). Porcine proliferative enteropathy (PPE) is a ubiquitous disease in pig-keeping countries around the world, with producers and veterinarians surveyed in the UK and Australia estimating that clinical symptoms have spread to 30–56% of herds [7,8]. In the same place, serological surveys of fattening pigs show that 80–100% of the herds are infected with *Lawsonia intracellularis* [9,10,11]. In Germany, Denmark, Spain, the Netherlands, and the United Kingdom, at least one positive sample was found in 90.3% (79.2–100%) of 144 herds investigated by quantitative PCR [12]. Máté et al. [13] also found 100% infection in 27 pig farms in Hungary. The pathogen of the disease is *Lawsonia intracellularis*, an obligate intracellular bacterium. The disease occurs not only in pigs but also in other species (e.g., horses) [14].

Clinically and sub-clinically ill animals excrete the pathogen in the feces to different extents. The infection occurs orally. The first histological changes in the small intestine are detected 8–10 days after infection. The pathological symptoms depend on which of the degenerative and reparative processes following the infection predominates.

The most common age group to be infected ranges from 6 weeks of age (young nursery pigs) to the end of fattening (20 weeks). The clinical symptoms are mild; it is also common to have detectable lesions only after slaughter. Average daily weight gain and feed conversion ratio decreases among diseased pigs. Sick animals are anorexic and occasionally have diarrhea. Mortality can be high.

Diagnosis of the disease is possible based on pathological and histopathological lesions. Immunohistochemistry or PCR may be required for causal diagnosis. Confirmation of PPE diagnosis is typically based on the observation of characteristic microscopic lesions. Difficulties in routine cultivation of *Lawsonia intracellularis* have led to the use of several alternative methods to detect infection, such as immunohistochemistry (IHC) or, less commonly, in situ hybridization. Immunohistochemical staining was the gold standard, with special use of the *Lawsonia intracellularis* specific antibody to identify the bacterium in PE lesions [15]. The presence of *Lawsonia intracellularis* can be confirmed by electron microscopy as a typical curved intracellular organism.

CR assays have been widely used to confirm *Lawsonia intracellularis* in feces or intestinal mucosal samples [16]. The susceptibility of PCR techniques to *Lawsonia intracellularis* has increased significantly, especially in the last decade. Many PCR assays can detect up to 100 organisms per gram of feces.

Due to this high sensitivity, the presence of the pathogen in the feces does not necessarily mean clinical importance in a given herd. When evaluating a negative IHC result, the distribution of PPE lesions in the intestinal tract should also be considered and it may be necessary to examine several segments to confirm the presence of the pathogen.

Sampling protocols should consider the occurrence of clinical signs (diarrhea, poorer weight gain, feed intake) in different age groups.

Antibiotics (e.g., tetracyclines, tiamulin, tylosine) are commonly used to treat or pre-vent PPE, and it is common practice to treat groups of animals continuously for extended periods. There are no data available to suggest that *Lawsonia intracellularis* develops resistance to commonly used antibiotics [17].

A few years ago, a vaccine appeared on the market that had to be administered orally. The results obtained with this vaccine were reliable with accurate dosing, but the method of application made it difficult, in many cases impossible, to apply it to large-scale pig farms. There has been a definite need to develop and market a precisely designed parenteral vaccine that provides safe administration. The recognition of the significant economic damage of the disease, its worldwide spread, the urgent need for prudent antibiotic use, and the possibility of a new vaccination strategy have created the need to develop an immediate and inexpensive manner of evaluating the impact of *Lawsonia intracellularis*, e.g., by monitoring pigs entering the slaughterhouse.

Jensen et al. (1997) first reported the monitoring of porcine enteric disease [4] in a slaughterhouse similar to mycoplasmosis. Their studies examined the ileal sections of 39 fattening pigs slaughtered at the slaughterhouse. Their visual and tactile monitor method suggested the following classifications of the examined ileum:

0 = normal proportions, rigidity, and thickness of the intestinal wall;

1 = rigidity and thickness of the intestinal wall increased slightly;

2 = intestinal wall thickened and rigidity up to 4 mm, lymphoid tissue hyperplasia,

hyperemia, and mild edema in the mucosa;

3 = thickening greater than 4 mm in the intestinal wall with partial stenosis.

The results of their studies show that it is possible to monitor the intestines by visual and palpation aspects after slaughter, and that the intestines can be classified into groups that can be checked by histopathological examinations based on the rigidity and thickness of the ileum. The intestines at the abattoir were macroscopically characterized for PPE-associated regional ileitis: hyperemia, sub serosal edema, and increased thickness of the rigid intestinal wall. However, other intestinal disorders may result in macroscopically similar results, so it is essential to demonstrate typical proliferation or to detect *Lawsonia intracellularis* bacteria. The increased wall thickness of the rigid intestines did not suggest any active infection, as the lesions were characterized by chronic, nonspecific, or late reactions to enteritis: hypertrophy of the tunica muscularis, Peyer’s patches, goblet cell hyperplasia, infiltration with mononuclear cells. *Lawsonia intracellularis* was detected by PCR in eight guts with increased thickness and one with a grade 0 test result. Of the nine PCR-positive intestines, three were also immunohistochemically positive and their feces were also found to be PCR positive. In two of the immunohistochemically positive intestines, *Lawsonia intracellularis* was detected only in macrophages.

In Hungary, in a slaughterhouse owned by a major pig integrator, it was found that the number of small intestines suitable for further processing decreased significantly (by almost 60–70%) during the production process due to inflammation, rupture and significant narrowing of the small intestine. Histological examinations confirmed the intracellular presence of *Lawsonia intracellularis* in several cases of samples showing macroscopic lesions. Vaccination against ileitis in farms producing slaughter pigs has significantly reduced the amount of small intestine that cannot be processed, while at the same time leading to a significant improvement in the use of antibiotics in that farm [5].

The essence of the method we have developed is that, similar to the scoring of the lungs, it can be performed during slaughterhouse processing so that it does not interfere with or make it impossible to carry out normal technological processes. The method can be applied without reducing the speed of the slaughtering line. The results of the procedure are correlated with the results of other laboratory diagnostic tests for ileitis (histology, immunohistochemistry, herd serology, fecal PCR). It is noteworthy that the results of the method can be used to immediately estimate the impact of *Lawsonia intracellularis* infection on the performance of the herd from which the slaughter animals come. Based on studies performed on a total of 27 swine units with nearly 850 fattening pigs, the method is suitable for achieving all the set objectives. Its introduction opens a new opportunity to estimate the losses caused by the disease, but also to regularly check and continuously monitor the practical effectiveness of therapies and preventive vaccination-based methods. The studies will allow the development of site-specific strategies (at what age to be vaccinated) adapted to the ileitis status of the herds. Regular slaughterhouse monitoring may also provide a basis for a total or partial reduction in antibiotic use (e.g., discontinuation of antibiotic use in feed).

The developed method uses the experience gained from previous studies with low animal numbers (Jensen et al. (1997) [4]) as well as the results of estimating the losses caused by ileitis (Holtkamp 2019 [6]). We are convinced that, with the introduction of the method, the same decisive tool will be put into practice in the hands of veterinarians and animal keepers as the examination of lung scoring.

As we had even less practice in the initial phase of applying the method, the difference in the use of the visual and tactile methods was mostly due to this. Overall, we believe that the visual and tactile method, with some practice, leads to the same result under practical conditions as the results of mucosal and intestinal examination after intestinal dissection. Therefore, in the following, we consider it sufficient to use the visual and tactile method. This is also particularly important in fulfilling the requirement that the test does not cause extra contamination during the normal cutting process.

## 5. Conclusions

The method developed is similar to lung lesion scoring tools and can be performed at slaughterhouses. The results of the procedure are correlated with the results of other laboratory diagnostic tests for ileitis (histology, immunohistochemistry, serology, fecal PCR). The results can provide an immediate interpretation of the status of ileitis.

Knowing these data, using the results of the slaughtered visual and tactile examination at the slaughterhouse, the magnitude of the loss caused by *Lawsonia intracellularis* infection can be estimated quickly and accurately, and the return on investment of the strategy to be applied can be accurately planned.

## Figures and Tables

**Table 1 animals-13-00542-t001:** VP versus mucosa, Hungary.

Farm	Type of the Farm	Lawsonia Vaccine User ?	Total Investigated Animals	Scoring of Visual Inspection and Palpation Method				Scoring of Intestine Mucosa and Content				Is There Preliminary Serological Test at Farm Level?	Fecal PCR Results			
Country: Hungary				0	1	2	3	0	1	2	3		Negative	Below Limit of Detection	*Lawsonia* Detected	10 log/µL
A	breeding	No	53	9	29	15		5	37	11		Yes	0	0	7	1.1–2.5
B	fattening	No	50	12	20	18		12	22	16		No	6	0	1	1.4
C	breeding	No	40	11	22	7		11	22	7		Yes	0	0	10	1.5–3.3
D	breeding	No	27	8	12	7		9	13	5		Yes	0	0	5	2.5–5.2
E	fattening	No	25	7	14	4		6	15	4		No	0	0	3	5.2–7.4
F	fattening	No	32	4	13	15		2	17	13		Yes	0	0	5	1–3.3
G	fattening	No	25	7	10	6	2	2	11	10	2	No	0	0	2	2–7
H	fattening	No	50	16	17	17		13	23	14		No	0	1	2	1.4–1.5
I	breeding	No	27	8	12	7		9	11	7		Yes	1	0	2	1.4
J	fattening	No	27	10	13	4		4	19	4		Yes	0	0	4	2.5–3.4
K	breeding	No	27	7	13	7		6	15	6		Yes	0	0	5	2.4–3.2
L	breeding	No	55	10	33	12		4	37	14		Yes	5	4	1	1
M	breeding	Yes	25	15	9	1		4	15	6		Yes	n.i.	n.i.	n.i.	
TOTAL			463	124	217	120	2	87	257	117	2					
in %			100%	27%	47%	26%	0%	19%	56%	25%	0%					

Note: n.i.: not investigated.

**Table 2 animals-13-00542-t002:** VP versus mucosa in RO, PL, CR, SL.

Country	Farm	Type of the Farm	Lawsonia Vaccine User ?	Total Investigated Animals	Scoring of Visual Inspection and Palpation Method				Scoring of Intestine Mucosa and Content				Is There Preliminary Serological Test at Farm Level?
					0	1	2	3	0	1	2	3	
Romania	A	breeding	No	25	3	15	6	1	n.i.	n.i.	n.i.	n.i.	Yes
	B	breeding	No	25	13	11	1	0	n.i.	n.i.	n.i.	n.i.	Yes
	C	breeding	Yes	30	26	4	0	0	n.i.	n.i.	n.i.	n.i.	Yes
	D	breeding	No	31	11	15	4	1	n.i.	n.i.	n.i.	n.i.	Yes
	Total			111	53	45	11	2					
				100%	48%	41%	10%	2%					
Poland	E	fattening	No	25	6	17	2	0	n.i.	n.i.	n.i.	n.i.	Yes
	F	fattening	No	32	9	21	2	0	n.i.	n.i.	n.i.	n.i.	Yes
	G	fattening	No	25	9	10	6	0	n.i.	n.i.	n.i.	n.i.	Yes
	H	fattening	No	20	5	10	5	0	n.i.	n.i.	n.i.	n.i.	Yes
	Total			102	29	58	15	0					
				100%	28%	57%	15%	0%					
Croatia	I	breeding	No	26	7	13	6	0	8	12	6	0	Yes
	J	breeding	No	25	11	10	4	0	12	10	3	0	Yes
	K	breeding	No	29	12	9	5	3	12	9	4	3	Yes
	Total			80	30	32	15	3	33	31	13	3	
				100%	38%	40%	19%	4%	41%	39%	16%	4%	
Slovakia	L	breeding	No	25	9	14	2	0	9	14	2	0	Yes
	M	breeding	No	25	8	14	3	0	8	14	3	0	Yes
	N	breeding	No	40	22	11	7	0	23	10	7	0	Yes
	Total			90	39	39	12	0	40	38	12	0	
				100%	43%	43%	13%	0%	44%	42%	13%	0%	
				383	151	174	53	5	73	69	25	3	
				100%	39%	45%	14%	1%	43%	41%	15%	2%	

Note: n.i.: not investigated.

**Table 3 animals-13-00542-t003:** VP vs. Mucosa significance.

Hungary	ENTERIPIG Score of Palpation at Slaughterhouses	ENTERIPIG Score of Mucosal Membrane at Slaughterhouses	%-Agree =	*p*-Value <
	≥1	≥1	84.9	0.001
	≥1	≥2	50.5	0.001
	≥2	≥1	45	0.001
	≥2	≥2	89.9	0.001
Croatia, Slovakia	ENTERIPIG Score of Palpation at Slaughterhouses	ENTERIPIG Score of Mucosal Membrane at Slaughterhouses	%-Agree =	*p*-Value <
	≥1	≥1	97.6	0.001
	≥1	≥2	57.1	0.001
	≥2	≥1	60.6	0.001
	≥2	≥2	98.8	0.001

**Table 4 animals-13-00542-t004:** VP versus Histopathology and IHA.

Score at Slaughterhouse	No of Samples	Histopathological and Immunohistochemical Findings (Hungarian Farms)																							
		Exfoliation		%	Inflammatory Cell (Lymphocytic, Histiocytic, Granulocytic) Infiltration		%	Crypt Epithelial Cell Proliferation		%	Edema		%	Hemorrhage		%	Lymphoid Hyperplasia		%	Fibrosis		%	Lawsonia Intracellularis		%
0	46	0	20	43%	0	28	61%	“+”	1	2%	“+”	2	4%	“+”	0	0%	“+”	42	91%	“+”	0	0%	“+”	1	2%
		1	23	50%	1	13	28%	“−”	45	98%	“−”	44	96%	“−”	46	100%	“−”	4	9%	“−”	46	100%	“−”	45	98%
		2	2	4%	2	4	9%																		
		3	1	2%	3	1	2%																		
1	62	0	8	13%	0	6	10%	“+”	7	11%	“+”	5	8%	“+”	0	0%	“+”	59	95%	“+”	0	0%	“+”	6	10%
		1	34	55%	1	28	45%	“−”	55	89%	“−”	57	92%	“−”	62	100%	“−”	3	5%	“−”	62	100%	“−”	56	90%
		2	17	27%	2	21	34%																		
		3	3	5%	3	7	11%																		
2	82	0	0	0%	0	0	0%	“+”	12	15%	“+”	17	21%	“+”	4	5%	“+”	81	99%	“+”	0	0%	“+”	12	15%
		1	11	13%	1	17	21%	“−”	70	85%	“−”	65	79%	“−”	78	95%	“−”	1	1%	“−”	82	100%	“−”	70	85%
		2	60	73%	2	44	54%																		
		3	11	13%	3	21	26%																		

**Table 5 animals-13-00542-t005:** Cohen’s analyses of VP and histopathology.

Hungary					
ENTERIPIG Palpation at Slaughterhouses	Histopathology Exfoliation		%-agree =	*p*-value <	
≥1	≥1		80.5	0.001	
≥1	≥2		68.9	0.001	
≥2	≥1		56.3	0.001	
≥2	≥2		80.5	0.001	
ENTERIPIG Palpation at slaughterhouses	Histopathology inflammatory cell infiltration		%-agree =	*p*-value <	
≥1	≥1		85.5	0.001	
≥1	≥2		68.4	0.001	
≥2	≥1		59.5	0.001	
≥2	≥2		74.7	0.001	
ENTERIPIG Palpation at slaughterhouses	Histopathology crypt epithelial cell proliferation		%-agree =	*p* value <	
≥1	neg	poz	60.5	0.077	n.s.
≥2	neg	poz	35.5	0.24	n.s.
ENTERIPIG Palpation at slaughterhouses	Histopathology Edema		%-agree =	*p*-value <	
≥1	neg	poz	62.6	0.076	n.s.
≥2	neg	poz	35.3	0.111	n.s.
ENTERIPIG Palpation at slaughterhouses	Histopathology lymphoid hyperplasia		%-agree =	*p*-value <	
≥1	neg	poz	46.3	0.243	n.s.
≥2	neg	poz	75.8	0.003	
ENTERIPIG Palpation at slaughterhouses	LAWSONIA_IC IH	LAWSONIA_IC IH	%-agree =	*p*-value <	
≥1	neg	poz	34.7	0.03	n.s.
≥2	neg	poz	61.1	0.001	

Note: n.s.: no significance.

**Table 6 animals-13-00542-t006:** VP versus Histopathology and IHA.

		Exfoliation		%	Inflammatory Cell (Lymphocytic, Histiocytic, Granulocytic) Infiltration		%	Crypt Epithelial Cell Proliferation		%	Edema		%	Hemorrhage		%	Lymphoid Hyperplasia		%	Fibrosis		%	Lawsonia Intracellularis		%
0	6	0	6	100%	0	6	100%	“+”	0	0%	“+”	0	0%	“+”	0	0%	“+”	0	0%	“+”	0	0%	“+”	0	0%
		1	0	0%	1	0	0%	“−”	6	100%	“−”	6	100%	“−”	6	100%	“−”	6	100%	“−”	6	100%	“−”	6	100%
		2	0	0%	2	0	0%																		
		3	0	0%	3	0	0%																		
1	42	0	23	55%	0	22	52%	“+”	0	0%	“+”	0	0%	“+”	0	0%	“+”	0	0%	“+”	0	0%	“+”	0	0%
		1	13	31%	1	16	38%	“−”	42	100%	“−”	42	100%	“−”	42	100%	“−”	42	100%	“−”	42	100%	“−”	42	100%
		2	6	14%	2	4	10%																		
		3	0	0%	3	0	0%																		
2	44	0	12	27%	0	11	25%	“+”	3	7%	“+”	0	0%	“+”	0	0%	“+”	0	0%	“+”	0	0%	“+”	0	0%
		1	14	32%	1	19	43%	“−”	41	93%	“−”	44	100%	“−”	44	100%	“−”	44	100%	“−”	44	100%	“−”	44	100%
		2	18	41%	2	13	30%																		
		3	0	0%	3	1	2%																		

**Table 7 animals-13-00542-t007:** Cohen’s analyses of VP and histopathology.

Romania, Poland, Croatia, Slovakia				
ENTERIPIG Palpation at Slaughterhouse	Histopathology Exfoliation		%-agree =	*p*-value <
≥1	≥1		63.0	0.004
≥1	≥2		32	0.13 (n.s.)
≥2	≥1		65.2	0.003
≥2	≥2		65.2	0.002
ENTERIPIG Palpation at slaughterhouse	Histopathology inflammatory cell infiltration		%-agree =	*p*-value <
≥1	≥1		25.0	0.007
≥1	≥2		42.4	0.001
≥2	≥1		65.2	0.003
≥2	≥2		65.2	0.002
ENTERIPIG Palpation at slaughterhouse	LAWSONIA_IC IH	LAWSONIA_IC IH	%-agree =	*p*-value <
≥1	neg	pos	9.78	0.64 (n.s.)
≥2	neg	pos	55.4	0.066 (n.s.)

Note: n.s.: no significance.

**Table 8 animals-13-00542-t008:** Serology, PCR, and VP method.

Country	Farm	Seroconversion	IH			PCR					Slaughterhouse			
		week	negative	doupt	poz	negative	1	2	3	more	0	1	2	3
Hungary	A	18	10	0	0	3	2	2	0	0	17%	55%	28%	0%
	B	n.a.	5	0	0	4	1	0	0	0	24%	40%	36%	0%
	C	13	13	1	1	0	2	6	3	0	27%	55%	18%	0%
	D	21	8	0	2	0	0	2	1	2	30%	44%	26%	0%
	E	12	10	0	0	0	2	1	1	1	12%	41%	47%	0%
	F	n.a.	2	0	8	0	0	0	0	3	28%	56%	16%	0%
	G	n.a.	8	0	7	0	0	1	0	1	28%	40%	24%	8%
	H	n.a.	15	0	0	3	2	0	0	0	32%	34%	34%	0%
	I	13	15	0	0	1	2	0	0	0	30%	44%	26%	0%
	J	16	10	0	0	0	0	3	1	0	37%	48%	15%	0%
	K	9	10	0	0	0	0	3	2	0	26%	48%	26%	0%
	L	7	36	0	0	9	1	0	0	0	18%	60%	22%	0%
	M (vaccine user)	12	n.i.			n.i.					60%	36%	4%	0%

Note: n.i.: non investigated.

**Table 9 animals-13-00542-t009:** Economical losses, Hungary.

	Herd Lawsonia Status by Holtkamp	Lawsonia Non-Affected		FL	5968.8	HUF	16.58	EUR	78.95	RON	75.11	PLN		
		Lower Bound		LB	4075.2	HUF	11.32	EUR	53.90	RON	51.28	PLN		
		Upper Bound		UB	601.2	HUF	1.67	EUR	7.95	RON	7.57	PLN		
		Mortality		MO	52390.8	HUF	145.53	EUR	693	RON	659.25	PLN		
		Visual and Palpation at Abattoir (ENTERIPIG)						FL	LB	UB	MO	HUF		
Hungarian Herds	No of Slaughtered Animal	0	1	2	3	HUF	5968.8	4075.2	601.2	52390.8	Realized Income	Profit compared to Lawsonia Non-Affected Status	Missed Profit/Slaughtered Pig	Realized Profit/Slaughtered Pig
A	100	17%	55%	28%	0%	596,880	101,469.6	224,136	16,833.6	0	342,439.2	57%	2544.4	3424.4
B	100	24%	40%	36%	0%	596,880	143,251.2	163,008	21,643.2	0	327,902.4	55%	2689.8	3279.0
C	100	27%	55%	18%	0%	596,880	161,157.6	224,136	10,821.6	0	396,115.2	66%	2007.6	3961.2
D	100	30%	44%	26%	0%	596,880	179,064	179,308.8	15,631.2	0	374,004	63%	2228.8	3740.0
E	100	12%	41%	47%	0%	596,880	71,625.6	167,083.2	28,256.4	0	266,965.2	45%	3299.1	2669.7
F	100	28%	56%	16%	0%	596,880	167,126.4	228,211.2	9619.2	0	404,956.8	68%	1919.2	4049.6
G	100	28%	40%	24%	8%	596,880	167,126.4	163,008	14,428.8	−419,126	−74,563.2	−12%	6714.4	−745.6
H	100	32%	34%	34%	0%	596,880	191,001.6	138,556.8	20,440.8	0	349,999.2	59%	2468.8	3500.0
I	100	30%	44%	26%	0%	596,880	17,9064	179,308.8	15,631.2	0	374,004	63%	2228.8	3740.0
J	100	37%	48%	15%	0%	596,880	220,845.6	195,609.6	9018	0	425,473.2	71%	1714.1	4254.7
K	100	26%	48%	26%	0%	596,880	155,188.8	195,609.6	15,631.2	0	366,429.6	61%	2304.5	3664.3
L	100	18%	60%	22%	0%	596,880	107,438.4	244,512	13,226.4	0	365,176.8	61%	2317.0	3651.8
M	100	60%	36%	4%	0%	596,880	358,128	146,707.2	2404.8	0	507,240	85%	896.4	5072.4

**Table 10 animals-13-00542-t010:** Economic losses in RO, PL, CR, SL.

		Herd Lawsonia Status by Holtkamp	Lawsonia Non-Affected		FL	5968.8	HUF	16.58	EUR	78.95	RON	75.11	PLN		
			Lower Bound		LB	4075.2	HUF	11.32	EUR	53.90	RON	51.28	PLN		
			Upper Bound		UB	601.2	HUF	1.67	EUR	7.95	RON	7.57	PLN		
			Mortality		MO	52390.8	HUF	145.53	EUR	693	RON	659.25	PLN		
			Visual and Palpation at Abattoir (ENTERIPIG)					FL	LB	UB	MO	EUR			
Country	Herd	No of Slaughtered Animals	0	1	2	3	EUR	16.58	11.32	1.67	145.53	Realized Income	Profit Compared to Lawsonia Non-Affected Status	Missed Profit/Slaughtered Pig	Realized Profit/Slaughtered Pig
Romania	A	100	12%	60%	24%	4%	1658	199	679	40	582	336	20%	13.2	3.4
	B	100	52%	44%	4%	0%	1658	862	498	7	0	1367	82%	2.9	13.7
	C	100	87%	13%	0%	0%	1658	1437	151	0	0	1588	96%	0.7	15.9
	D	100	35%	48%	13%	3%	1658	588	548	22	469	688	42%	9.7	6.9
Poland	E	100	24%	68%	8%	0%	1658	398	770	13	0	1181	71%	4.8	11.8
	F	100	28%	66%	6%	0%	1658	466	743	10	0	1220	74%	4.4	12.2
	G	100	36%	40%	24%	0%	1658	597	453	40	0	1090	66%	5.7	10.9
	H	100	25%	50%	25%	0%	1658	415	566	42	0	1022	62%	6.4	10.2
Croatia	I	100	27%	50%	23%	0%	1658	446	566	39	0	1051	63%	6.1	10.5
	J	100	44%	40%	16%	0%	1658	730	453	27	0	1209	73%	4.5	12.1
	K	100	41%	31%	17%	10%	1658	686	351	29	1505	−439	−26%	21.0	−4.4
Slovakia	L	100	36%	56%	8%	0%	1658	597	634	13	0	1244	75%	4.1	12.4
	M	100	32%	56%	12%	0%	1658	531	634	20	0	1185	71%	4.7	11.8
	N	100	55%	28%	18%	0%	1658	912	311	29	0	1252	76%	4.1	12.5

## Data Availability

Detailed data, pictures and notes are available via the corresponding author’s e-mail address.

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
