# Peer review of "Slaughterhouse Visual and Palpation Method for Estimating the Economic Damage of Porcine Proliferative Enteropathy (PPE)"

_animals, 2023, doi:10.3390/ani13030542_

Round 1
Reviewer 1 Report
Dear authors,
The manuscript is well-written and has a high interest to the swine industry due to the practical application in the area of intestinal disease management and control in relation to Lawsonia intracellularis.
I have detected some minor elements that should be addressed in order to further improve the manuscript:
L16-17 - The sentence should be re-edited: 'Ileitis is one of the most important and widely spread enteric diseases.'
L41 and further - Lawsonia intracellularis should be written in italic. This is the case at several locations throughout the manuscript. Please have a thorough check.
L90 - Please homogenize the terminology 'farrow-to-slaughter' in the same way through the manuscript. Sometimes, it is written with '-' sometimes without.
L112 and further - The term 'formula' is used at different points in the materials and methods. However, to the reader, it might not be too clear what this means. Please adapt to get full insight into what is the content of this term.
L166 - I think that the correct expression of 1000 kg of feed is a 'tonne'. Please adapt this in the text on the different locations where it is used.
Table 1. Please adapt the width of the columns (especially column 2 / 3 / 4) in order to get the full words on the same line and not things like 'fattenin g'. There should be sufficient space within the table to get thing moved in a correct way to achieve this.
Table 1. Would it be possible to get everything on one page? Especially taking into account the preceeding remark!
L209 - Please indicate 'Table 3 - continued'.
Table 5 - Please move the table to get it entirely on one page.
Table 6 - Add a capital letter to 'slaughterhouse' in the table headings.
Table 7 - part 2 - Please add title 'Table 7 - continued' above.
L297 - There a two '. .' at the end of the sentence.
Table 8 - Please exchange 'abattoir' with 'slaughterhouse' since this is used throughout the text.
Table 8 - Please adapt column width to get everything better fitted into the table.
L325 - Shouldn't it be 'Clinically and subclinically ill ...'.
L331-332 - Remove the 'enter' and get the sentence follow on the same line. This is to my opinion no break and it disturbs the readability of the sentence.
L339 - I would recommend to fuse the sentences to the following suggestion: '... alternative methods to detect infection, such as immunohistochemistry (IHC) or, ...'
Author Response
Dear Reviewer No 1.
We would like to thank you for the accurate, detailed, well-founded, and justified corrections that we have accepted and included in the manuscript.
Please see the attachment.

Reviewer 2 Report
line 99 - perhaps do not refer to "us" - refer to "the methods described in this manuscript"
line 112 - the reviewer does not know what "hard tact formula" means - please use a different description
similar comments throughout the description of the scoring system - please redefine "formula" as a term that is recognized universally
ileum mucosa scoring system - are "lymph follicles" referring to "lymph nodes"?
line 190 is a bit awkward as written - please consider rewriting
Table 1 - define "VP vs mucosa". The title needs to describe the table as a stand alone entity in the manuscript
line 199-200 - similar comment for line 190
line 201 "lack of a comfortable place" - for collection of samples? please provide a more descriptive explanation for not making mucosa investigations
similar comment on Table 2 - abbreviations need to be fully defined in the table title or with a legend
line 237 - why is it labeled Table 1/a?
why is table 4/b so much smaller than the other table representing similar information?
the tables are a bit difficult to interpret - please revise titles and define terms in them - the reviewer is unsure of what some of the numbers in the columns represent.
line 332 needs to be on line 331
Author Response
Dear Reviewer No2.,
We would like to thank you for the accurate, detailed, well-founded, and justified corrections that we have accepted and included in the thesis.
Please see the attachment
